# Dissipation-Driven Selection under Finite Diffusion: Hints from Equilibrium and Separation of Time Scales

**DOI:** 10.3390/e23081068

**Published:** 2021-08-17

**Authors:** Shiling Liang, Paolo De Los Rios, Daniel Maria Busiello

**Affiliations:** 1Institute of Physics, School of Basic Sciences, École Polytechnique Fédérale de Lausanne—EPFL, 1015 Lausanne, Switzerland; shiling.liang@epfl.ch (S.L.); paolo.delosrios@epfl.ch (P.D.L.R.); 2Institute of Bioengineering, School of Basic Sciences, École Polytechnique Fédérale de Lausanne—EPFL, 1015 Lausanne, Switzerland

**Keywords:** reaction networks, non-equilibrium systems, diffusion

## Abstract

When exposed to a thermal gradient, reaction networks can convert thermal energy into the chemical selection of states that would be unfavourable at equilibrium. The kinetics of reaction paths, and thus how fast they dissipate available energy, might be dominant in dictating the stationary populations of all chemical states out of equilibrium. This phenomenology has been theoretically explored mainly in the infinite diffusion limit. Here, we show that the regime in which the diffusion rate is finite, and also slower than some chemical reactions, might bring about interesting features, such as the maximisation of selection or the switch of the selected state at stationarity. We introduce a framework, rooted in a time-scale separation analysis, which is able to capture leading non-equilibrium features using only equilibrium arguments under well-defined conditions. In particular, it is possible to identify fast-dissipation sub-networks of reactions whose Boltzmann equilibrium dominates the steady-state of the entire system as a whole. Finally, we also show that the dissipated heat (and so the entropy production) can be estimated, under some approximations, through the heat capacity of fast-dissipation sub-networks. This work provides a tool to develop an intuitive equilibrium-based grasp on complex non-isothermal reaction networks, which are important paradigms to understand the emergence of complex structures from basic building blocks.

## 1. Introduction

Any chemical system in non-equilibrium conditions with time-independent transition rates will eventually reach a non-equilibrium stationary state [1]. This is maintained at the expenses of a constant energy consumption and manifests into the presence of steady currents. Predictions stemming from equilibrium arguments about the abundance of chemical species often dramatically fail when there are external sources of energy [2,3]. Indeed, it has been recently shown that non-equilibrium conditions can trigger stabilisation effects in molecular and chemical systems [4,5,6,7]. Additionally, the fact that in out-of-equilibrium regimes kinetic aspects are usually as relevant as non-dissipative diffusive properties has been investigated in recent years [8,9,10].

Recent works [11,12] have studied the consequences of applying a thermal gradient to a diffusive chemical system. In particular, they elucidated that non-equilibrium conditions couple with an underlying kinetic asymmetry in the transition rates, favouring, at stationarity, a subset of chemical states that are unfavourable at equilibrium [13]. Inspired by the idea that complex, high-energy states could have been populated at the dawn of life in non-equilibrium conditions, they refer to this asymmetry in the steady occupation probability of chemical states as selection. Its strength between any two pair of states can be quantified by the unbalance of their steady probabilities. Moreover, this emergent phenomenon is associated with specific features of the stationary energy dissipation into the environment [14]. In a nutshell, non-isothermal conditions allow states that dissipate energy faster, i.e., those participating to the faster reaction pathways and named fast-dissipation states, to be more populated than states with a slower rate of dissipation, at stationarity.

All the presented results about the non-equilibrium selection of states are valid in regimes where the Arrhenius law is applicable [1,15]. Here, we consider again this setting, and the modifications to chemical rates arising from kinetic theory and time-scales analyses lie beyond the subject of this work [16,17]. However, the major limitation of previous findings on this topic resides in the ideal assumption of a spatial diffusion much faster than all other processes in play, even if they are qualitatively valid even outside this limit.

Here, we explore more realistic cases in which the diffusion coefficient between two thermal reservoirs at different temperatures is finite, and the diffusive time scale is comparable to the chemical transition rates [18]. Interestingly, as a function of the diffusion coefficient, the system may experience sharp transitions between phases with different selection strengths. We also find that it is possible to achieve higher selection than in the fast diffusion limit, as well as an inversion in the state that will be selected at stationarity. This complex picture can be captured by a time-scale separation analysis under some approximations, and, as a consequence, we find that appropriate local equilibrium predictions can give precious hints to rephrase and understand these non-equilibrium behaviours.

## 2. Phase-Transition for Selection in Two-State Systems

In order to fix the ideas, we first consider the simplest case of two chemical species diffusing between two boxes at different temperatures [11]. $P(X_{i})$ is the time-dependent probability to be in the state *X* (X=A,B), within the box *i* (i=1,2), where i=1 indicates the cold box, while i=2 refers to the hot box. Hence, $P(X_{i})$ satisfies the following reaction–diffusion equations [1]:(1)$$\begin{array}{c} \partial_{t}P\left(A_{i}\right)=-k_{B_{i}A_{i}}P\left(A_{i}\right)+k_{A_{i}B_{i}}P\left(B_{i}\right)+d_{A}\left(P\left(A_{j}\right)-P\left(A_{i}\right)\right)\\ \partial_{t}P\left(B_{i}\right)=+k_{B_{i}A_{i}}P\left(A_{i}\right)-k_{A_{i}B_{i}}P\left(B_{i}\right)+d_{B}\left(P\left(B_{j}\right)-P\left(B_{i}\right)\right) \end{array}$$
where i=1,j=2 or vice versa; thus, the set ${\left\{A_{i},B_{i}\right\}}_{i=1,2}$ identifies all the possible chemical states of the system in both boxes. Here, $d_{A},d_{B}$ are the diffusion coefficients of species A,B, respectively, and $k_{XY}$ indicates the transition rate from state *Y* to state *X*. When representing a thermally activated transition, as in this case, $k_{XY}$ has the following Arrhenius form [19,20,21,22]:(2)$$\begin{array}{c} k_{A_{i}B_{i}}=k_{0}\exp\left(-\frac{∆E+\varepsilon }{T_{i}}\right)\;k_{B_{i}A_{i}}=k_{0}\exp\left(-\frac{\varepsilon }{T_{i}}\right)\; \end{array}$$
where $T_{i}$ is the temperature of box *i*, ∆E the energy difference between the two chemical states, ϵ the energetic barrier, and $k_{0}$ a constant pre-factor.

It has been observed [23] that when $d_{A}\neq d_{B}$, thermophoresis [24,25] can occur, and particles accumulate on one side of the gradient. However, this effect is not detrimental to the selection of chemical states, and we consider $d_{A}=d_{B}=d$ throughout the whole manuscript unless stated otherwise. Moreover, here, we consider $T_{1}\equiv T_{c}<T_{2}=T_{c}+∆T\equiv T_{w}$. The temperature gradient injects energy into the system and triggers the onset of a non-equilibrium stationary state. Thermal energy is converted into chemical energy, in the form of an unbalance in state occupancies [11].

In this case, we cannot define a chemical selection, since we have only one low-energy state, *B*; hence, no kinetic symmetry breaking is possible under non-equilibrium conditions [11], since there could not be any kinetic asymmetry [13]. Indeed, to trigger a selection, the minimal ingredient is the presence of two possible reactions from the high-energy state, one fast and one slow, towards two different low-energy states. Which one of these two will be selected at stationarity is dictated by their kinetics, along with their energies, in out-of-equilibrium conditions [11]. However, in the present case, we consider as a relevant observable the ratio between the total population of the species *B*, $\bar{P_{B}}=P\left(B_{1}\right)+P\left(B_{2}\right)$, and *A*, $\bar{P_{A}}=P\left(A_{1}\right)+P\left(A_{2}\right)$, that is, $R_{BA}=\bar{P_{B}}/\bar{P_{A}}$. From now on, we use the symbol $\bar{\;\cdot \;}$ to indicate the sum of · in both boxes.

As shown in Figure 1a, when the transport coefficient *d* is finite, we observe a sharp transition in the behaviour of $R_{BA}$ around a critical barrier, $\varepsilon_{d}$. As *d* increases, for any given value of the energetic barrier, $R_{BA}$ decreases. In the limit of fast diffusion, the transition disappears, hinting at the non-trivial role of finite diffusion in chemical diffusive systems. In order to have an intuitive estimation of $\varepsilon_{d}$, we consider the time scales of the processes in play. The diffusion is determined by the rate *d*, while chemical reactions have their own rates, with the cold box supporting slower reactions. The sharp transition has to occur when these time scales become comparable. Hence, $\varepsilon_{d}$ is defined as the barrier satisfying the following equation:(3)$$k_{0}\exp\left(-\varepsilon_{d}/T_{c}\right)=d\;\to \;\varepsilon_{d}=-T_{C}\ln\frac{d}{k_{0}}$$

Naively speaking, the slowest downhill transition, i.e., the one in the cold box that populates *B*, is equal to the diffusion of *B* between boxes, when $\varepsilon =\varepsilon_{d}$. We will later generalise this argument on a more firm ground, based on a time-scale separation procedure. Nevertheless, despite handwaving, the estimate of $\varepsilon_{d}$ is compatible with numerical simulations (see Figure 1c).

As a function of the energetic barrier, we can identify three different behaviours (see Figure 1a: (i) When $\varepsilon <\varepsilon_{d}$, the system is in the fast-dissipation regime. The system will relax within each box before diffusing so that the steady state, in this limiting case, is given by the average of two Boltzmann distributions—one at temperature $T_{c}$, and the other at temperature $T_{w}$ [1]. This is also the maximum possible value for $R_{BA}$ (see Appendix A); (ii) the transition regime, in which $\varepsilon \approx \varepsilon_{d}$; (iii) the slow-dissipation regime, when $\varepsilon >\varepsilon_{d}$. In this case, all reactions in the cold box are much slower than diffusion and do not contribute to determining the stationary state. This coincides with the Boltzmann distribution at temperature $T_{w}$. It is evident, even in this simple setting, that equilibrium distributions, along with considerations about the interplay among time scales, might provide useful information about the system’s behaviours in different genuinely out-of-equilibrium conditions. We also remark that non-equilibrium effects still remain visible in the microscopic fluxes circulating in the system [11,12]. As shown in Figure 1b, particles store energy in the hot box, populating *A*, then diffuse to the cold side, release heat in the cold box, populating *B*, and finally diffuse back to restart the thermal cycle.

We stress that for two-state systems, the infinite diffusion case gives lower values of $\bar{P_{B}}/\bar{P_{A}}$ as follows:(4)$$\frac{\bar{k_{A\to B}}}{\bar{k_{B\to A}}}=\frac{k_{A_{1}\to B_{1}}+k_{A_{2}\to B_{2}}}{k_{B_{1}\to A_{1}}+k_{B_{2}\to A_{2}}}$$
as already derived in [11].

## 3. Simplest Case for Selection: A Three-State System

Bearing in mind the complex picture described so far, here we investigate a three-state system, which is the simplest case in which it is possible to introduce a kinetic asymmetry and hence define a selection. Again, we consider the presence of one high-energy state, *A*, that can convert into two low-energy states, *B* and *C*, with the same energy (for the sake of simplicity). The energy barrier between *A* and *B*, $\varepsilon_{B}$, is lower than the one between *A* and *C*, $\varepsilon_{C}$. $∆\epsilon =\varepsilon_{C}-\varepsilon_{B}$ quantifies the kinetic asymmetry (see Figure 2a). At equilibrium, *B* and *C* end up being equally populated, since they have the same energy. When exposed to a temperature gradient, the system exhibits a selection: the fast-dissipation state, *C*, has a higher population than the slow-dissipation state *B*, in the infinite diffusion limit [11,12]. The selection parameter is $R_{CB}=\bar{P_{C}}/\bar{P_{B}}$.

Without delving deeper into this model, which are extensively reported in [11], we explore what happens when the diffusive rate *d* is finite. As reported in Figure 2b,c, the infinite diffusion case does not always lead to the optimal selection. In the numerical example reported in Figure 2b, after reducing the value of *d*, the selection reaches a maximum value which is sensibly higher than the one obtained for d→+∞. However, Figure 2c reports a situation in which finite *d* leads to $R_{CB}$ lower than the one obtained in the infinite diffusion limit.

We follow the same reasoning of the previous section to understand this outcome. Consider the case of Figure 2b. Here, the energy landscape is nearly flat, i.e., $∆E\ll \epsilon_{C},\epsilon_{B}$; hence, the limiting chemical rates (in the cold box) are dominated by the energetic barrier. When d→0, all reactions are much faster than diffusion and the system returns to equilibrium in each box, $R_{CB}=0$. Increasing *d*, the reaction between *A* and *B* in the cold box starts becoming slower than diffusion, while the one between *A* and *C* stays faster, because of the kinetic asymmetry. Naively speaking, *C* equilibrates between both boxes, and the transformation between *A* and *B* can be ignored for the steady state, resulting in a positive stationary $R_{CB}$. Increasing *d*, in this case, also the reaction between *A* and *C* in the cold box becomes slower than diffusion, and the system falls back into the infinite diffusion limit [11,18,23]. When the energy landscape is not flat, i.e., $∆E\approx \epsilon_{C},\epsilon_{B}$, chemical reactions are not governed solely by energetic barriers, and the system is more complicated to analyse on intuitive basis.

Again, we remark that, at least in the case of a nearly flat energy landscape, considerations about time scales and properly derived equilibrium solutions might improve our understanding of this (slightly more complete) chemical non-equilibrium system.

## 4. Time-Scale Separation and Equilibrium Hints

### 4.1. Fast-Dissipation Chemical Sub-Networks in Two-Box Models

The time scale associated with a chemical reaction is the inverse of its corresponding transition rate. This quantity also dictates the dissipation speed along a specific reaction pathway. Analogously, the time scale associated with diffusion is 1/d, which is also the average occupation time of each box. Intuitively, as discussed above, fast transitions tend to equilibrate the system in their sub-space, providing a reliable approximation of the non-equilibrium steady state, employing only equilibrium solutions in fast-dissipation sub-spaces.

To make these observations quantitative and elucidate their limits, we here build a time-scale separation analysis for a generic chemical network [18,26]. First, we consider the simple case of a nearly flat energy landscape, $∆E\ll \varepsilon_{B},\varepsilon_{C}$. In this case, we define two classes of transitions—slow and fast. Slow transitions are associated with the characteristic time $\tau_{S}$, while fast reactions act on the time scale $\tau_{F}$. Consider, for example, a slow transition from the state *i* to *j* happening at temperature *T* as follows:(5)$$k_{ji}^{S}=k_{0}^{S}e^{-\frac{∆E+\varepsilon_{ij}^{S}}{T}}\approx k_{0}^{S}e^{-\frac{\varepsilon_{ij}^{S}}{T}}=k_{ij}^{S}=\kappa_{ij}\tau_{S}^{-1}$$
where $\kappa_{ij}$ are reaction-specific deviations from a given slow average transition rate $1/\tau_{S}$.

The condition of nearly flat energy landscape is manifestly crucial to ensure that each transition is slow or fast independently of the direction, i.e., ∆E does not play a determinant role. As a consequence, we can split the transition matrix determining the evolution of the system, $\hat{K}$, in two parts, ${\hat{K}}^{S}$ and ${\hat{K}}^{F}$, respectively, containing only slow and fast transitions. Indeed, the (ij)-th element of ${\hat{K}}^{S}$ is $k_{ji}^{S}$, for i≠j, while the diagonal elements are $k_{ii}^{S}=-\sum_{j\neq i}k_{ji}^{S}$ in order to have a normalised probability. Further, $k_{ij}^{S}$ can be written as in Equation (Equation 5). All these observations hold analogously for ${\hat{K}}^{F}$.

In a two-box model, such as the one described above, we have the following dynamics:(6)$$\begin{array}{cc} \partial_{\tau }P\left(X_{1}\right)= & \sum_{Y_{1}\neq X_{1}}\left(\kappa_{X_{1}Y_{1}}^{S}P\left(Y_{1}\right)-\kappa_{Y_{1}X_{1}}^{S}P\left(X_{1}\right)\right)\\ & +\frac{\tau_{S}}{\tau_{F}}\sum_{Z_{1}\neq X_{1}}\left(\kappa_{X_{1}Z_{1}}^{F}P\left(Z_{1}\right)-\kappa_{Z_{1}X_{1}}^{F}P\left(X_{1}\right)\right)+d\tau_{S}\left(P\left(X_{2}\right)-P\left(X_{1}\right)\right) \end{array}$$
where X,Y,Z=A,B,C,… indicate the chemical state, while the subscripts 1 and 2 represents the box. A similar equation holds for $P(X_{2})$. In the model here considered, ${\hat{K}}^{F}$ includes the totality of reactions in the hot box and only a fraction of them in the cold box. Here, $\tau =t/\tau_{S}$ is a slowly evolving a-dimensional time.

First, when $\tau_{F}^{-1}>\tau_{S}^{-1}\gg d$, the system follows a Boltzmann equilibrium distribution in both the hot and cold box, with temperature $T_{2}\equiv T_{w}$ and $T_{1}\equiv T_{c}$, respectively, as for the d→0 case [11,18]. Indeed, the fast-dissipation sub-spaces are the chemical reaction networks in each box.

Conversely, the other limiting case already studied in [11,18] is $d\gg \tau_{F}^{-1}>\tau_{S}^{-1}$. The system falls back into the fast-diffusion limit, and the stationary state for the total probability is
(7)$$\lim_{d\to +\infty }P\left(X_{i}\right)=\frac{1}{2}\Pi^{\mathrm{st}}\left(X\right)$$
where $\Pi^{\mathrm{st}}\left(X\right)$ is the stationary distribution of a chemical network with effective rates ${\tilde{k}}_{XY}=\bar{k_{XY}}$, living in one single box. Here, $\Pi^{\mathrm{st}}\left(X\right)$ differs in general from an equilibrium solution, since the effective rates have no longer the Arrhenius form. In this case, the fast-dissipation sub-network is composed only of diffusive links. Hence, the system first equilibrates in this sub-space, reaching a spatial equilibrium distribution, which is uniform in space. Indeed, the numerical factor in Equation (Equation 7) derives from the fact that the probabilities within each box are normalised to 1/2 in this regime. Therefore, the population is distributed among chemical states according to an *effective* (a-spatial) equilibrium.

Finally, when $\tau_{F}^{-1}\gg d\gg \tau_{S}^{-1}$, we are in the richer situation of a finite diffusion regime, with three different time scales in play. We propose a solution to Equation (Equation 15) of the following form:(8)$$P\left(X_{i}\right)=P^{\left(0\right)}\left(X_{i}\right)+\frac{\tau_{F}}{\tau_{S}}P^{\left(1\right)}\left(X_{i}\right)$$

Before proceeding further, we remark that fast-dissipative sub-networks are, in general, disconnected sets of chemical reactions, $S_{1},\ldots ,S_{N}$, in the cold box. On the contrary, since $T_{w}$ is associated with faster reactions, the whole chemical system in the hot box is a fast-dissipative sub-network. Hence, at the zeroth order in $\tau_{F}/\tau_{S}$, we have a set of disconnected master equations, one for each $S_{i}$, and also a master equation governing the dynamics in the hot box:(9)$$\begin{array}{cc} 0 & =\sum_{Y_{1}^{\left(i\right)}}\left(\kappa_{X_{1}^{\left(i\right)}Y_{1}^{\left(i\right)}}^{F}P^{\left(0\right)}\left(Y_{1}^{\left(i\right)}\right)-\kappa_{Y_{1}^{\left(i\right)}X_{1}^{\left(i\right)}}^{F}P^{\left(0\right)}\left(X_{1}^{\left(i\right)}\right)\right)\;\forall i=1,\ldots N\;\mathrm{subnetworks}\mathrm{cold}\mathrm{box}\\ 0 & =\sum_{Y_{2}}\left(\kappa_{X_{2}Y_{2}}^{F}P^{\left(0\right)}\left(Y_{2}\right)-\kappa_{Y_{2}X_{2}}^{F}P^{\left(0\right)}\left(X_{2}\right)\right)\;\mathrm{whole}\mathrm{network}\mathrm{hot}\mathrm{box}. \end{array}$$

In the first line, the superscript (i) indicates states belonging to the *i*-th sub-network in the cold box.

Equation (Equation 9) is solved by the generic form $P^{\left(0\right)}\left(X_{i}\right)=p\left(i\right)\Pi^{F}\left(X_{i}\right)$. Here, $\Pi^{F}\left(X_{i}\right)$ is the solution for the state $X_{i}$, satisfying the chemical master equation to which $X_{i}$ belongs. The pre-factor p(i) depends only on the box considered, and it cannot be determined from the zeroth order. Indeed, inserting the expression for $P^{\left(0\right)}$ back, summing over all chemical states, and solving the system up to the first order in $d\tau_{F}\gg \tau_{F}/\tau_{S}$, we determine p(i) by
(10)0=p(2)−p(1)p(1)=p(2)=1/2

This ensures that the probability of finding a particle in box 1 or 2 is equally distributed due to diffusion.

Note that $P^{\left(0\right)}$ satisfies only equilibrium equations at the leading order since the diffusion enters only in Equation (Equation 10). Hence, equilibrium solutions of fast-dissipative sub-networks are sufficient to provide an accurate approximation of the complete solution in the presence of a net separation of time scales.

Here, the role of the hot box is to excite the chemical reaction network moving particles towards the high energy state. This excitation can also be achieved through other mechanisms, such as photon absorption. One can extract the characteristic time scales of these excitations, compare them with chemical reaction rates, and build a similar analysis based on fast-dissipative sub-spaces. Again, reaction networks might be decomposed into fast-dissipative sub-nets, with the steady-state distribution resulting as a composition of equilibrium distributions of sub-spaces.

### 4.2. Fast-Dissipation Ensemble Distribution in Two-Box Models

Fast-dissipative sub-networks, however, exhibit interesting ensemble properties that have to be taken into account in order to provide a complete solution to the system in terms of equilibrium distributions.

In fact, considering the first-order solution to the full dynamics, and summing over all chemical states constituting each sub-network, $\left(ij\right)\in S_{z}$, and using Equation (Equation 10), we obtain the following:(11)$$P\left(S_{i}|T_{c}\right)=P\left(S_{i}|T_{w}\right)=\frac{1}{2Z_{w}}\sum_{z\in S_{i}}e^{-\frac{E_{z}}{k_{B}T_{w}}}$$
where $E_{z}$ is the energy of the state *z*, $Z_{w}$ is the partition function of the hot box, so that $P(S_{i}|T_{w})$ is the probability to be in the fast-dissipating set $S_{i}$ in the hot box at the leading order. Analogously, $P(S_{i}|T_{c})$ is the probability of occupation of $S_{i}$ in the cold box at the leading order. Note that $S_{i}$ is a fast-dissipative sub-network only in the cold box (see Equation (Equation 9)). Here, $Z_{h}$ is a normalisation factor. The last equality derives from the fact that high temperature is associated with faster reactions, and then the hot box supports reactions always faster than diffusion in this context. Hence, the ensemble of fast-dissipative sub-networks in the cold box follows an equilibrium-like distribution. Moreover,
(12)$$P\left(X\in S_{i}|T_{c}\right)=P\left(S_{i}|T_{c}\right)\frac{1}{Z_{c}\left(S_{i}\right)}e^{-\frac{E_{X}}{k_{B}T_{c}}}$$
where $Z_{c}\left(S_{i}\right)$ is the partition function relative to the sub-network $S_{i}$ in the cold box.

There also exist situations in which the diffusion rate from the cold to the hot box, $d_{c\to w}$ is not equal to the reverse one, $d_{w\to c}$, for example for geometrical reasons. In this case:(13)$$P\left(S_{i}|T_{c}\right)=\frac{d_{c\to w}}{d_{w\to c}+d_{c\to w}}\frac{1}{Z_{w}}\sum_{z\in S_{i}}e^{-\frac{E_{z}}{k_{B}T_{w}}}$$

The interest in these equilibrium-like solutions is twofold: they can provide an intuitive understanding of the interplay between non-equilibrium conditions and selection, and they also can be verified experimentally, without complete knowledge of the whole system.

### 4.3. Numerical Results and Energy Landscapes

In Figure 3a, we report a two-box model for a complex system with a nearly flat energy landscape. Solid lines identify fast-dissipation reactions, whose transition rates are greater than diffusion. Conversely, dashed lines are transitions slower than diffusion, which can be ignored to find the stationary solution at a first order, as shown above. We also highlight with circles the fast-dissipative sub-networks in the cold box, whose equilibrium solutions (considering only fast reactions) provide valuable information about the complete steady-state distribution. It is evident that all reactions in the hot box are marked with solid lines so that the entire chemical network at temperature $T_{w}$ constitutes a fast-dissipative sub-network. The transition between different regimes is controlled by the value of the diffusion coefficient.

Figure 3b,c presents the agreement between the exact stationary probabilities, obtained numerically integrating the master equation describing the system and the theoretical predictions stemming from the proposed method, Equation (Equation 12). In particular, in Figure 3b, we show the Kullback–Leibler divergence (*K*) between exact and theoretical solutions, which acts as a global estimator of their similarity, as a function of the variance of all state energies, an indicator of the roughness of the landscape. Indicating, for the sake of simplicity, with $P^{\mathrm{exact}}\left(X\right)$ and $P^{\mathrm{theory}}\left(X\right)$, respectively, the exact and theoretical steady-state probability to be in the state *X*, we have
(14)$$K=\sum_{X}P^{\mathrm{exact}}\left(X\right)\log\frac{P^{\mathrm{exact}}\left(X\right)}{P^{\mathrm{theory}}\left(X\right)}$$
where the sum runs over all states. Moreover, in Figure 3c, we present three specific cases of increasing roughness of the energy landscape, from left to right. For flat or not extremely complex energy landscapes (left and central panels), we stress that the agreement is remarkable, even beyond the strict limits of our theoretical derivation, while deviations appear for rough landscapes (right panel). Here, we set only two values for the energetic barriers—one for fast and the other for slow reactions—in order to have a net separation of time scales, and then we modified the energy landscape. Another possibility, which gives analogous results, is to draw transition rates from a given distribution. Clearly, when there is no net separation of time scales, our method cannot be straightforwardly applied.

### 4.4. Fast-Dissipation Chemical Sub-Networks for Continuous Systems

In order to complete the discussion, we here consider the presence of a continuous thermal gradient T(x) and describe the system through the probability to be in a given state *i*, at a given time *t* and position *x*, $P_{i}\left(x,t\right)$. In this case, diffusive reactions are replaced by the diffusion Laplacian operator. Clearly, this setting generalises the two-box model, and the evolution reads as follows:(15)$$\begin{array}{c} \partial_{\tau }P_{i}\left(x,t\right)=\sum_{j\neq i}\left(\kappa_{ij}^{S}P_{j}\left(x,t\right)-\kappa_{ji}^{S}P_{i}\left(x,t\right)\right)\\ +\frac{\tau_{S}}{\tau_{F}}\sum_{n\neq i}\left(\kappa_{in}^{F}P_{n}\left(x,t\right)-\kappa_{ni}^{F}P_{i}\left(x,t\right)\right)+D\tau_{S}\partial_{x}^{2}P_{i}\left(x,t\right) \end{array}$$
where *D* has the dimension of a diffusion coefficient.

When $\tau_{F}^{-1}\gg d\gg \tau_{S}^{-1}$, we are in the finite diffusion regime. The main difference with respect to the two-box model is that here we cannot distinguish time scales of reactions according to the temperature T(x) at which they occur. In fact, since the temperature varies continuously over the entire domain, sub-networks could be different for each point in space. On the flip side, if some energetic barriers are high enough so that independently of T(x) the reactions associated with them will be slower than all the others in play, we can distinguish two different time scales as before, and consequently identify some fast-dissipation sub-networks. Hence, we propose the following solution to Equation (Equation 15):(16)$$P_{i}\left(x,t\right)=P_{i}^{\left(0\right)}\left(x,t\right)+\frac{\tau_{F}}{\tau_{S}}P_{i}^{\left(1\right)}\left(x,t\right)$$

The resulting zeroth-order equations have the following form:(17)$$0=\sum_{j\in S_{z}}\left(\kappa_{ij}^{F}\left(x\right)P_{j}^{\left(0\right)}\left(x,t\right)-\kappa_{ji}^{F}\left(x\right)P_{i}^{\left(0\right)}\left(x,t\right)\right)\;\forall z=1,\ldots N,$$
where *N* is the number of disconnected sub-networks, as before. The general solution is $P_{i}^{\left(0\right)}\left(x,t\right)=p\left(x\right)\Pi_{i}^{F}\left(x\right)$, where $\Pi_{i}^{F}\left(x\right)$ is the equilibrium solution of the fast chemical sub-network to which *i* belongs. Here, p(x) encodes spatial variations, and it cannot be determined from the zeroth-order equations. Solving the system up to the first order in $d\tau_{F}$, by summing over all chemical states, we determine p(x) as the solution of the following diffusive equation:(18)$$0=\partial_{x}^{2}P\left(x\right)$$

Again, due to diffusion, the solution is homogeneously distributed in space, strengthening the parallel between two-box models and continuous-space systems in this presented framework.

## 5. Diffusion-Controlled Switch of Selection

Chemical systems, in general, are composed of a set of low-energy states with different energies. Hence, at equilibrium, states are selected according to those, following the Boltzmann distribution. There are situations [12] in which this energetic selection is in competition with the kinetic non-equilibrium selection [13], when fast reactions do not drive the system towards the lowest energy state.

Since the value of the diffusion coefficient, *d*, controls the strength of kinetic (dissipation-driven) selection, the existence of these two competing mechanisms can cause a switch of the selected state at stationarity as a function of *d*. In Figure 4, we show this effect for a paradigmatic three-state, two-box model. For small *d*, i.e., $d<k_{A\to B}\left(T_{c}\right)$, the reactions in both boxes are in local equilibrium so that the state *B* is more populated than *C* in the steady state,
(19)$$\begin{array}{cc} P_{B}\; & =\frac{e^{-E_{B}/k_{B}T_{c}}}{2Z(T_{c})}+\frac{e^{-E_{B}/k_{B}T_{w}}}{2Z(T_{w})}>\frac{e^{-E_{C}/k_{B}T_{c}}}{2Z(T_{c})}+\frac{e^{-E_{C}/k_{B}T_{w}}}{2Z(T_{w})}=P_{C}. \end{array}$$

When the diffusion coefficient takes intermediate finite values, $k_{A\to B}\left(T_{c}\right)=e^{-\varepsilon_{B}/k_{B}T_{c}}<d<k_{A\to C}\left(T_{c}\right)=e^{-\varepsilon_{C}/k_{B}T_{c}}$, the transition A⇋C constitutes a fast-dissipation reaction, while A⇋B supports slower dissipation. Hence, *A* and *C* form a fast-dissipative sub-network reaching local equilibrium in both boxes. However, state *B* can not be reached from state *A* in the cold box and thus stays in equilibrium with its high-temperature counterpart to which it is connected by diffusion. Here, combining Equations (Equation 9)–(Equation 11), P(B), and P(C) are determined by the following equilibrium system:(20)$$\begin{array}{ccc} & \; & \;\;\mathrm{Equilibrium}\mathrm{in}\mathrm{the}\mathrm{Hot}\mathrm{Box}\\ \dot{P}\left(A_{2}\right) & = & -k_{C_{2}A_{2}}P\left(A_{2}\right)+k_{A_{2}C_{2}}P\left(C_{2}\right)-k_{B_{2}A_{2}}P\left(A_{2}\right)+k_{A_{2}B_{2}}P\left(B_{2}\right)=0\\ \dot{P}\left(B_{2}\right) & = & -k_{A_{2}B_{2}}P\left(B_{2}\right)+k_{B_{2}A_{2}}P\left(A_{2}\right)=0 \end{array}$$(21)$$\begin{array}{ccc} & \; & \;A⇋C\;\mathrm{equilibrium}\mathrm{in}\mathrm{the}\mathrm{Cold}\mathrm{Box}\\ \dot{P}\left(A_{1}\right) & = & -k_{C_{1}A_{1}}P\left(A_{1}\right)+k_{A_{1}C_{1}}P\left(C_{1}\right)=0 \end{array}$$(22)$$\begin{array}{ccc} & \; & \;\;\;\;\;\mathrm{Constrains}\mathrm{by}\mathrm{diffusion}\\ P(B_{1}) & = & P(B_{2})\\ 0.5 & = & P\left(A_{1}\right)+P\left(C_{1}\right)+P\left(B_{1}\right)\\ 0.5 & = & P\left(A_{2}\right)+P\left(B_{2}\right)+P\left(C_{2}\right) \end{array}$$

In the example reported, a strong dissipation-driven selection dominates in this regime (see Figure 4).

Further increasing *d*, all reactions become slower than diffusion, and the whole system is effectively dominated by energetic selection in the following infinite-diffusion limit:(23)$$R_{BC}=\frac{\bar{P_{B}}}{\bar{P_{C}}}=\frac{\bar{k_{C\to A}}\;\bar{k_{A\to B}}}{\bar{k_{A\to C}}\;\bar{k_{B\to A}}}>1$$

The exploration of regimes of finite diffusion, along with the theoretical framework here developed, which is based on equilibrium solutions, might lead to novel phenomena markedly different from those observed in the small- and large-diffusion limit. The existence of a dissipation-controlled switch of selection elucidates this possibility, and this work takes a step towards an a priori understanding of the role of *d* with the minimum knowledge of the equilibrium distributions of fast-dissipative sub-networks. Indeed, thermodynamic equilibrium does not depend on energetic barriers, which, in the presented framework, only determine the range of validity of the theoretical predictions.

## 6. Equilibrium Hints for Entropy Production

Energy dissipation in discrete-state systems can be quantified using Schnakenberg entropy production, ${\dot{S}}_{\mathrm{tot}}$ [14,22]. This can be divided into two terms—one accounting for the entropy change of the system, ${\dot{S}}_{\mathrm{sys}}$, and the other associated with the heat dissipated into the environment, ${\dot{S}}_{\mathrm{env}}$. Since, by definition, ${\dot{S}}_{\mathrm{sys}}=\left(d/dt\right)\left(\sum_{i}P_{i}\ln P_{i}\right)$, it vanishes in the steady state. Hence, ${\dot{S}}_{\mathrm{tot}}={\dot{S}}_{\mathrm{env}}$, and it also quantifies the heat absorbed by the hot box or expelled into the cold box. Employing the energy conservation principle, we have
(24)$${\dot{S}}_{\mathrm{tot}}=\sum_{\left(ij\right)}\left(k_{ij}P_{j}-k_{ji}P_{i}\right)\ln\frac{k_{ij}}{k_{ji}}=d\sum_{X}E_{X}\left(P\left(X_{2}\right)-P\left(X_{1}\right)\right)\left(\frac{1}{T_{1}}-\frac{1}{T_{2}}\right)$$
where the first sum $\sum_{\left(ij\right)}$ runs over all the possible pairs of nodes, while the second sum $\sum_{X}$ runs over all states in each box. Substituting in Equation (Equation 24) the solutions of the master equation, $P^{\mathrm{exact}}\left(X\right)$ for the state *X*, we obtain the exact entropy production (red solid line in Figure 5a). However, it is possible to estimate energy dissipation in the framework developed so far, by imposing the fast equilibration of fast-dissipative sub-networks. Substituting the steady-state solutions obtained from Equation (Equation 12) in Equation (Equation 24), indicated above as $P^{\mathrm{theory}}\left(X\right)$ for simplicity, we derive a theoretically approximated version of the entropy production which is valid when the diffusion coefficient is in a desired intermediate range (see also Figure 4). For a simple three-state system, shown in Figure 5a, we indicate that the theoretical ${\dot{S}}_{\mathrm{tot}}$ (blue dashed line) exhibits an excellent agreement with the exact entropy production.

Moreover, the proposed approach allows us to identify fast-dissipative sub-networks in the cold box, satisfying Boltzmann equilibrium in their sub-spaces. Hence, it is possible to define, for each of them, the heat capacity, $C_{v}^{\mathrm{fast}}\left(S_{i}\right)=\partial_{T}{\langle E\rangle }_{S_{i}}^{\mathrm{fast}}$, where the average is taken over all the states belonging to $S_{i}$. Again, the fast-dissipative sub-network in the hot box coincides with the entire chemical network. When ∆T is small, the entropy production, non-zero only out of equilibrium, can be expressed in terms of $C_{v}^{\mathrm{fast}}\left(S_{i}\right)$, equilibrium quantities, as follows:(25)$${\dot{S}}_{\mathrm{tot}}\approx d∆T\left(\frac{1}{T_{1}}-\frac{1}{T_{2}}\right)\sum_{i}P\left(S_{i}|T_{c}\right)C_{v}^{\mathrm{fast}}\left(S_{i}\right)$$

In the expression above, $P(S_{i}|T_{w})$ does not appear because of the relation in Equation (Equation 11). In Figure 5a, we compare Equation (Equation 25) (yellow dot-dashed line) with the exact entropy production, reporting an excellent agreement for small ∆T.

Additionally, we split the entropy production in the contributions from slow- and fast-dissipation reactions, without using any equilibrium mapping, and we find that the dominant role is played by the reaction path A⇋C, in accordance with previous results (see Figure 5b).

## 7. Discussion and Conclusions

Another experimentally feasible way to introduce a temperature gradient is to put the chemical system in contact with a heat bath whose temperature is periodically changed over time. In several conditions, this turns out to be easier than applying a steady thermal gradient [27,28]. A recent work [11] shows that the time-integrated selection in the time-periodic steady state can be exactly mapped to the stationary selection for a two-box system. The same equivalence is shown in Figure 6 in the case of finite diffusion, strengthening the intimate connection between these two frameworks to set the system in out-of-equilibrium conditions.

To summarise, here, we presented a method to deal with complex reaction networks in non-equilibrium conditions triggered by temperature differences. This method is rooted in a time-scale separation analysis, which allows going beyond the infinite diffusion limit and the quasi-equilibrium approximation, capturing behaviours in finite diffusion regime. The power of the proposed approach is that, under some approximations, the genuine non-equilibrium steady state can be understood from equilibrium solutions of fast-dissipative sub-networks, which are also accountable for the vast majority of the entropy production in the system.

With this method, we also showed that the finite diffusion regime hides numerous intricacies and peculiarities, as a switch of the selected state, or a boost in the selection strength. It would be interesting to push forward the parallel between theoretical idealised systems and experimentally feasible procedures, in order to verify these theoretical predictions. Moreover, observing features of non-isothermal chemistry might also ignite the study of the origins of life problems from the point of view of non-equilibrium statistical mechanics and thermodynamics [29,30,31,32,33,34].

## Figures and Tables

**Figure 1 entropy-23-01068-f001:**
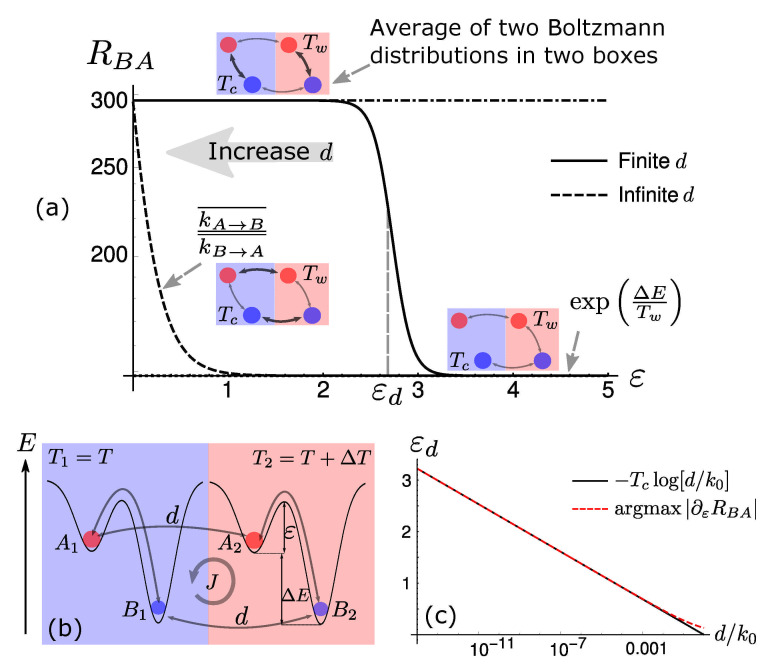
(**a**) $R_{BA}=\bar{P_{B}}/\bar{P_{A}}$ as a function of the energetic barrier ε for both finite and infinite diffusion cases. Small insets are sketches of the reaction network in different conditions, where the thickness of the arrows reflects the speed of the corresponding reaction. The solutions in these settings are also reported next to the insets. (**b**) Sketch of a two-state, two-box reaction network in a temperature gradient, where the inner circular arrow represents the direction of non-equilibrium stationary flux *J*. (**c**) The theoretical critical point $\varepsilon_{d}$ scales linearly with $d/k_{0}$, showing a clear agreement with numerical estimations, with small deviations only for large values of *d*.

**Figure 2 entropy-23-01068-f002:**
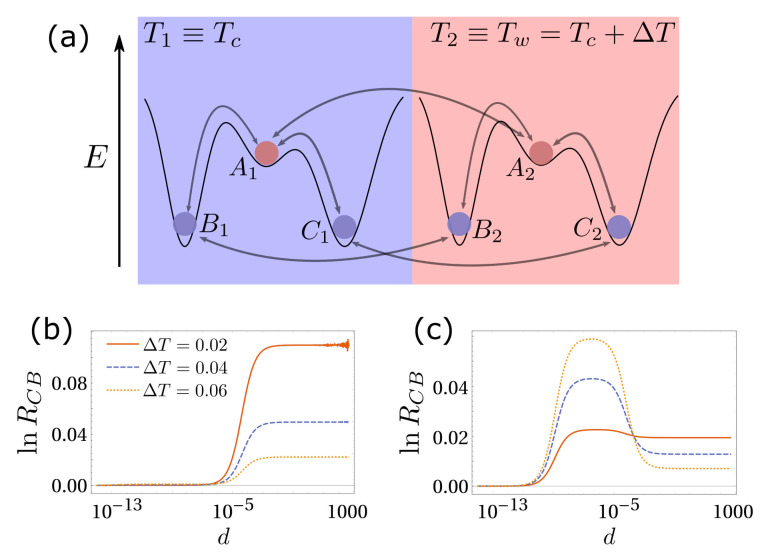
(**a**) Three-state, two-box reaction network. The same color corresponds to the same value of the energy. The energy difference between *A* and *B* (or *C*) is ∆E; (**b**) $∆E=1,T_{c}=0.1,\varepsilon_{C}=1,\varepsilon_{B}=2$. We report the logarithm of the selection parameter, $R_{CB}=\bar{P_{C}}/\bar{P_{B}}$, as a function of *d* for different values of the thermal gradient. $R_{CB}$ is maximised at infinite diffusion. (**c**) In this case, ∆E=0.1, while all other parameters and the color code are the same as in panel (**b**). Here, we show a peak in the selection strength for finite diffusion.

**Figure 3 entropy-23-01068-f003:**
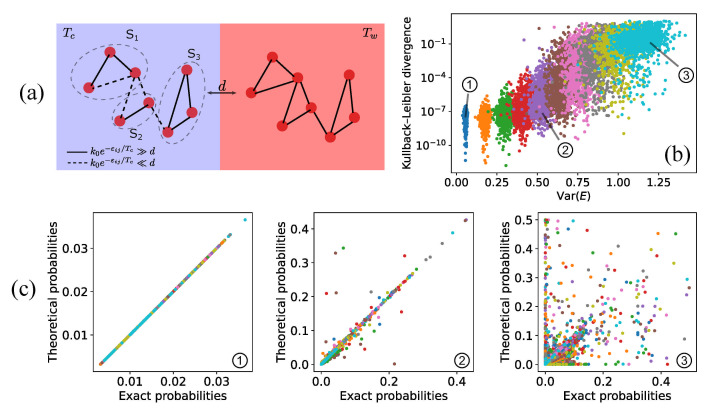
(**a**) Two-box model for a complex reaction network. Solid lines represent fast-dissipation reactions, while slow-dissipation reactions are indicated by dashed lines. Gray dashed circles indicate the fast-dissipative sub-networks in the cold box. The distribution inside each sub-network follows the Boltzmann equilibrium. Overall, the probability is redistributed by diffusion so that half of the total particles populate each box; (**b**) Kullback–Leibler divergence between exact solutions and theoretical predictions for the stationary probability distribution of different reaction networks composed of 40 states each. Networks are generated by randomly assigning fast-dissipation (ε=1) and slow-dissipation (ε=5) reactions between states that uniformly populate a given energy range, $R_{E}$. The Kullback–Leibler divergence is shown for $10^{3}$ reaction networks with the same energy range (identified by the same color), and for an increasing energy range (here estimated through the variance of the energies). As the roughness of the landscape increases, the proposed framework starts failing; (**c**) theoretical and exact probabilities in comparison for $R_{E}=1,2$, and 2.5 from left to right (here different colors represent different networks).

**Figure 4 entropy-23-01068-f004:**
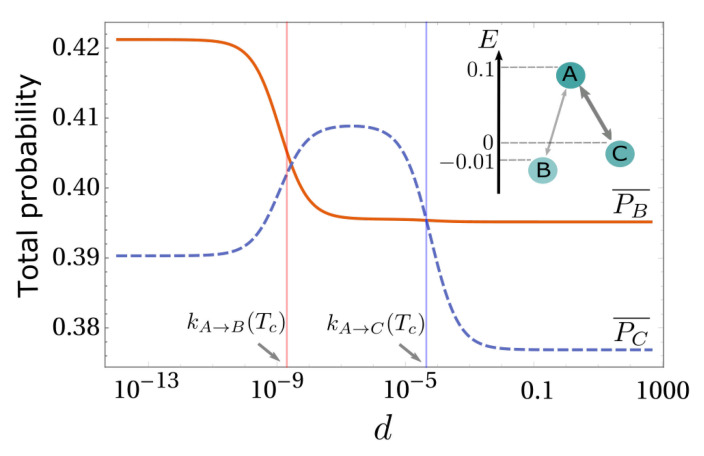
Total probability of states *B*, $\bar{P_{B}}=P\left(B_{1}\right)+P\left(B_{2}\right)$, and *C*, $\bar{P_{C}}=P\left(C_{1}\right)+P\left(C_{2}\right)$, as a function of *d*. The diffusion coefficient can trigger a switch of the selected state at stationarity, which is due to the competition between dissipation-driven and energetic selection. The diffusion coefficients at which each transition occurs can be estimated by comparison with the chemical reaction rate in the cold box. The upper inset sketch the chemical network here investigated. Parameters are $T_{c}=0.1$, $T_{w}=0.2$, $\varepsilon_{C}=1$, and $\varepsilon_{B}=2$.

**Figure 5 entropy-23-01068-f005:**
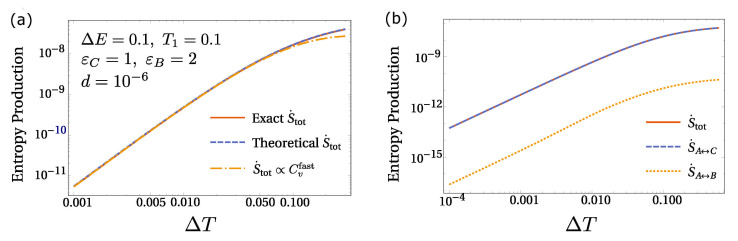
(**a**) For a simple three-state system, we compare the exact entropy production (solid red line) with the theoretical approximated one obtained using our framework (blue dashed line), and the formula obtained by a small gradient expansion, Equation (Equation 25) (yellow dot-dashed line). In the exact ${\dot{S}}_{\mathrm{tot}}$ the probabilities stem from direct solution of the master equation, while in the theoretical ${\dot{S}}_{\mathrm{tot}}$, and in Equation (Equation 25), the probabilities are obtained employing the equilibration of fast-dissipative sub-networks. To consistently apply our approach, we choose an intermediate value of the diffusion with respect to the chemical rates (see also Figure 4), showing an excellent agreement among the curves presented. (**b**) It is evident that the main contribution to the entropy production comes from the reaction A⇋C, ${\dot{S}}_{A↔C}$, which supports a much faster dissipation with respect to the slow-dissipation branch A⇋B. Indeed, we also see that ${\dot{S}}_{A↔B}\ll {\dot{S}}_{A↔C}$. Parameters are reported in panel (**a**).

**Figure 6 entropy-23-01068-f006:**
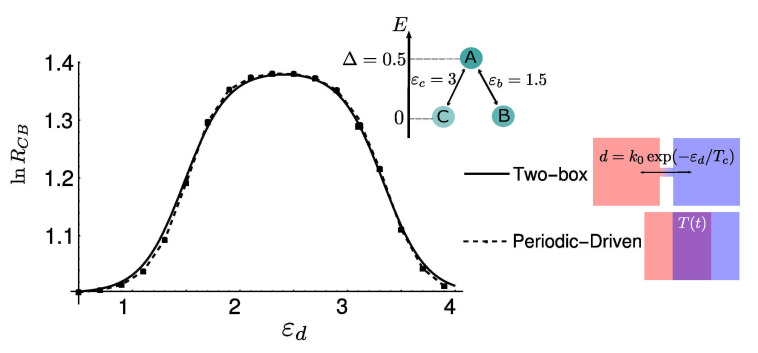
The logarithm of the selection strength as a function of the critical barrier $\varepsilon_{d}$ for a three-state, two-box model (solid line), as sketched in the inset, and for a time-periodic driven three-state system (dashed line). These two paradigms are qualitatively and quantitatively equivalent to determine non-isothermal selection of states.

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
