# Peer review of "Dissipation-Driven Selection under Finite Diffusion: Hints from Equilibrium and Separation of Time Scales"

_entropy, 2021, doi:10.3390/e23081068_

Round 1
Reviewer 1 Report
The manuscript “Dissipation-driven selection under finite diffusion: hints from equilibrium and separation of time-scales” reports interesting theoretical results on how diffusion in temperature fields can alter the frequency of states in a classical jump system. I believe that it provides inspiring material that should be published in Entropy. Below I report some suggestions for the Authors.
- The introduction is missing a definition of “selection”, “selection strength”, and of “fast/slow dissipation states”. Their meaning is gradually understood only during the reading of the text.
- Concerning population “inversion” of states, let me mention that some years ago these papers provided examples and a discussion on how kinetic aspects can be as relevant as dissipation, in nonequilibrium:
* Maes, Netočný and O'Kelly de Galway, Low temperature behavior of nonequilibrium multilevel systems. Journal of Physics A: Math. Theor. 47, 035002 (2014).
* Basu and Maes, Nonequilibrium Response and Frenesy. J. Phys.: Conf. Ser. 638, 012001 (2015).
There is also this recent interesting book:
* Christian Maes, Non-Dissipative Effects in Nonequilibrium Systems. SpringerBriefs in Complexity, ISBN 978-3-319-67780-4 (2018).
Since diffusion is a nondissipative effect, I believe these references my be of some relevance.
- About eq.(1), P is not defined in the text and it should also be specified that i = 1, j = 2 refers to the two subsystems at different T. Again, this emerges only by reading the text.
- What is the “kinetic symmetry breaking” of ref [10]?
- The usage of “R” for the ratio of probabilities should be started at line 63 and continued consistently in the rest of the text. Currently it is used intermittently or only in figures.
- Perhaps the factor ½ in (7) deserves a comment.
- Before eq.(8) there are typos: “more reacher”, and ref to later (14). See also “Indee” at line 230.
- I am wondering if Fig.3 could be complemented by a Kullback-Leibler divergence vs some meaningful estimator of the network features (“roughness”?). The KL would provide a summary for a whole network of how different are exact from theoretical probabilities.
- In Fig.5, there should be a clearer distinction of what is meant by exact and theoretical.
Author Response
We thank the referee for their valuable comments. Please find attached the response file.
Best regards,
Daniel M. Busiello, on behalf of all co-authors

Reviewer 2 Report
The manuscript studies chemical reactions in a system with finite diffusion rates. The method to describe complex reaction networks under non-equilibrium conditions caused by the temperature gradient is proposed based on a time-scale separation analysis. The method may help in understanding non-equilibrium steady-state starting from equilibrium solutions. The effect of diffusion rates on the probabilities of stationary states is evaluated. Some features of selection are discussed, such as a switch of selected state and a boost in the selection strength under a finite diffusion regime.
It is an interesting manuscript, well written and containing new original results. Nevertheless, there are several issues to be addressed before publication.
- The proposed approach is phenomenological and based on conceptual analysis rather than on rigorous theoretical formalism; at least, the style of presentation leads to such an impression. On the other hand, there is a number of studies (see, for instance, [1-4]) based on the strict kinetic theory of non-equilibrium chemical reactions under various kinetic scaling. They take into account various time scales for collisional processes and fluid dynamics; constitutive relations for diffusion velocities, heat flux, stress tensor, and reaction rates are derived from the Boltzmann equation; this allows evaluating the effect of temperature, pressure, velocity gradients on the reaction rates in real (not idealized) systems in a rigorous way. In particular, the effects of gas compressibility on the reaction rates, deviations from the Arrhenius law and the law of mass action caused by slow processes are discussed. The kinetic-theory approach with corresponding references has to be mentioned in the manuscript, and the advantages/limitations of the authors’ approach have to be emphasized.
- According to the Curie principle, thermodynamic forces/fluxes of various tensor dimensions are independent unless some electromagnetic fields are applied. The rate of chemical reaction is a scalar specified by the reaction affinity and, in a moving gas, by the velocity divergence, whereas the diffusion velocity is a vector specified by the diffusion driving force and temperature gradient. Therefore, the cross-coupling effects between chemistry and diffusion cannot be expressed explicitly from irreversible thermodynamics. This point has to be discussed.
- The study is parametric and uses non-dimensional values. However, it is not clear, to what specific conditions may correspond diffusion coefficients in the range 1e-13-1000; what is the meaning of \Delta T=0.02, etc. Some connection to real systems/conditions has to be provided.
- Is there any experimental confirmation for the mechanisms discussed in Section 3?
- Lines 194-195: “predictions obtained with the proposed method and exact numerical results”… How the exact results were obtained? Which equations are solved in the exact approach, and using which methods
- Define explicitly E_z and Z_w in Eq.(11).
References
[1] E. Nagnibeda, E. Kustova, Nonequilibrium Reacting Gas Flows, Kinetic Theory of Transport and Relaxation Processes, Springer Verlag, Berlin, Heidelberg, 2009.
[2] V. Giovangigli, Multicomponent Flow Modeling, Birkhauser, Boston, 1999.
[3] E. V. Kustova and D. Giordano, Cross-coupling effects in chemically non-equilibrium viscous compressible flows, Chem. Phys. 379(1–3), 83–91 (2011).
[4] E. G.Kolesnichenko and Yu. E. Gorbachev, Gas-dynamic equations for spatially inhomogeneous gas mixtures with internal degrees of freedom. I. General theory, Appl. Math. Modell. 34(12), 3778–3790 (2010).
Author Response

(The authors gave the same response as above.)

Round 2
Reviewer 2 Report
I am completely satisfied with the authors' response to my comments.